# Self-Guided Low Light Object Detection Framework

**Gwangik Shin , Jaeha Song , Soonmin Hwang**[†]
Department of Automotive Engineering
Hanyang University
Seoul, South Korea
`{david5432, archiiive99, soonminh}@hanyang.ac.kr`

## Abstract

Object detection in low-light environments is inherently challenging due to limited contrast and heavy noise, both of which significantly degrade feature representations. In this paper, we propose a novel self-guided low-light object detection framework that effectively addresses these issues without introducing additional parameters or increasing inference time. Our method incorporates a detachable auxiliary pipeline during training, consisting of an image enhancement module and a denoising module, followed by a Fourier-domain fusion block. This pipeline improves the feature representation of the detector's backbone, enhancing its robustness under low-light conditions. Importantly, at inference time, our method incurs no additional computational cost compared to the baseline detector while achieving substantial performance improvements. Extensive experiments on widely used low-light object detection benchmarks, such as DARK FACE and ExDark, demonstrate that our method achieves state-of-the-art performance. Notably, experiments on the nuImages dataset show that our approach can outperform domain adaptation methods—especially when a large domain gap between source and target domains is inevitable in the real-world applications—highlighting its practical effectiveness. Code is available at `https://github.com/gw-shin/SGLDet`.

## 1 Introduction

As autonomous driving technology is rapidly advancing, the robustness of object detection algorithms in low-light conditions has become increasingly critical for reliable environment perception and safe driving. While sensor-fusion approaches incorporating LiDAR or thermal cameras have been explored to handle nighttime conditions (Rashed et al., 2019; Lu & Lu, 2021), they inevitably incur additional hardware costs and computational overhead. Consequently, achieving effective low-light perception using only cameras remains both crucial and challenging. However, camera-based detection algorithms face inherent limitations, including excessive noise, blurred object boundaries, and distortions such as glare and reflections due to artificial light sources. These factors can significantly impair detection performance, thereby increasing the risk of nighttime traffic accidents. Enhancing object recognition under low-light environments has thus emerged as a key challenge in the development of autonomous driving systems.

Object detection in low-light scenarios faces several critical challenges. A common strategy is to apply low-light image enhancement (LLIE) techniques as a preprocessing step to improve visual quality (Chen Wei, 2018; Liu et al., 2021a; Ma et al., 2022a; Xu et al., 2020; Sun et al., 2022). However, these methods are primarily designed for human perception, often prioritizing visual appeal over detection utility, and thus contribute little to actual performance improvement. Alternatively, specialized low-light detectors have been proposed (Sasagawa & Nagahara, 2020; Cui et al., 2024; Peng et al., 2024), but they typically introduce high complexity and longer inference times. Explicit denoising, on the other hand, is rarely adopted during inference: while strong denoisers (Lee et al., 2022; Pan et al., 2023) can effectively suppress noise, they also blur structural details and incur

---

[†]Corresponding author.

heavy latency, making them unsuitable for real-time deployment. These limitations highlight the need for a new paradigm that can address noise while preserving both speed and accuracy.

To address these limitations, we propose a novel framework that integrates a self-guided mechanism into a baseline object detector, enhancing low-light detection without introducing additional inference overhead. Specifically, our approach utilizes only the input low-light image itself to estimate a supervisory target image that improves feature representation in the detector's backbone. The self-guided block addresses two key challenges in low-light scenarios: insufficient brightness and high noise. To tackle this issues, we adopt separate self-supervised modules for image enhancement and noise suppression, then merge their outcomes in the Fourier domain to produce a supervisory signal that preserves both structural and perceptual cues. This detachable auxiliary task, guided by the fused target, enables the baseline detector to adapt to the characteristics of low-light data. Importantly, all these processes are applied only during training, keeping the inference-time detector unchanged and incurring no additional computational cost. The main contributions are summarized as follows:

- We present a new framework design for low-light object detection that strategically employs detachable auxiliary supervision during training without inference-time computational overhead.
- We incorporate enhancement and denoising into low-light detection through Fourier-domain fusion that combines denoised amplitude components with enhanced phase components, allowing our framework to address noise while preserving structural details.
- Our framework leverages self-supervised modules that operate solely on low-light images, eliminating the requirement for paired ground-truth data and enabling adaptation to diverse nighttime environments.
- Experiments across multiple datasets and detectors demonstrate the effectiveness and generalizability of the proposed approach, showing significant performance improvements in different low-light detection scenarios.

## 2 RELATED WORK

### 2.1 LOW-LIGHT IMAGES

**Low-Light Image Enhancement** Deep learning-based Low-Light Image Enhancement (LLIE) has been explored as a crucial approach to improving visual quality. Many methods rely on supervised learning (Chen Wei, 2018; Chen et al., 2018; Zhang et al., 2019; Wu et al., 2022; Cai et al., 2023; Zhou et al., 2024), requiring paired low-light and ground-truth normal images for training. While these methods achieve high scores on image quality metrics, they are impractical in real-world outdoor scenes with dynamic objects, where ground-truth images are unavailable. To overcome this limitation, zero-reference methods have been introduced (Zhang et al., 2020; Jiang et al., 2021; Liu et al., 2021a; Li et al., 2021), incorporating non-reference loss functions without paired supervision such as estimating enhancement curves (Guo et al., 2020) or introducing a self-calibration for illumination estimation (Ma et al., 2022a). However, most LLIE methods focus more on visual quality rather than downstream vision tasks such as object detection. In low-light conditions, they often amplify noise or blur object boundaries, leading to performance degradation of detection models.

**Image Denoising** Image denoising is a crucial process for reducing high noise levels in low-light environments. Supervised methods train models using clean-noisy image pairs (Zhang et al., 2017; Anwar & Barnes, 2019; Yue et al., 2019; Kim et al., 2020; Yue et al., 2020; Liu et al., 2021b), but collecting such data across diverse scenes—especially in outdoor driving scenarios with dynamic objects—is impracticable. To address this, self-supervised methods (Krull et al., 2019; Huang et al., 2021; Wu et al., 2020; Lee et al., 2022; Pan et al., 2023; Chen et al., 2024) leverage Blind-Spot Networks (BSN) (Krull et al., 2019), enabling noise mitigation without paired supervision. AP-BSN (Lee et al., 2022) removes spatially correlated noise using pixel-shuffle downsampling. SDAP (Pan et al., 2023) incorporates inter-sample differences, and knowledge distillation is introduced to achieve high performance with lightweight networks (Chen et al., 2024). Nevertheless, most denoising approaches focus solely on noise reduction, often introducing blurring or high latency, which makes them unsuitable for object detection. This has led to denoising being rarely adopted in low-light detection, a gap our work specifically aims to address.

## 2.2 Object Detection in Low-light Conditions

One approach for low-light object detection is to employ a Low-Light Image Enhancement module (Chen Wei, 2018; Chen et al., 2018; Zhang et al., 2019; Wu et al., 2022; Guo et al., 2020; Liu et al., 2021a; Ma et al., 2022a; Cai et al., 2023; Zhou et al., 2024) as a preprocessing step. However, since LLIE methods are optimized for visual quality rather than object detection, their effectiveness in detection tasks is often limited. This limitation has led to jointly optimize the image processing pipeline with the detection task for better alignment (Hashmi et al., 2023; Sun et al., 2022; Ma et al., 2022b; Wang et al., 2022; Qin et al., 2022). Another approach focuses on modifying the detector architecture (Sasagawa & Nagahara, 2020; Hong et al., 2021; Cui et al., 2024; Peng et al., 2024) to improve performance under low-light conditions. These methods introduce multi-model merging (Sasagawa & Nagahara, 2020), feature enhancement modules (Peng et al., 2024), and illumination-aware representation learning (Cui et al., 2024). However, both approaches introduce additional computational overhead compared to the baseline detector, limiting their practicality for real-time applications.

Alternatively, domain adaptation techniques (Hong et al., 2021; Wang et al., 2021; Cui et al., 2021; Du et al., 2024) aim to transfer knowledge learned from a normal-light source domain to a low-light target domain. Various methods employ strategies such as multi-task autoencoding (Cui et al., 2021), low-light image synthesis (Hong et al., 2021), and Retinex-based adaptation (Du et al., 2024). However, their effectiveness heavily depends on the degree of similarity between the source and target domains. Although these methods also do not introduce additional inference overhead, their performance often deteriorates in scenarios with substantial domain gaps, such as nighttime driving, due to the source domain bias.

## 3 Method

### 3.1 Self-Guided Low-light Detection Framework

Existing approaches to low-light object detection commonly rely on complex image enhancement as a preprocessing step, customized detector architectures, or domain adaptation strategies. Enhancement-based methods often amplify noise and increase complexity at inference stage, modified detectors typically require architectural changes and additional effort during training, and domain adaptation techniques exhibit inconsistent performance across datasets due to varying domain gaps. In contrast, our approach directly uses only the original low-light image as input during inference, avoiding both information loss and extra computational cost. To achieve this, we introduce a *detachable auxiliary pipeline*, used only during training, which estimates a target image that serves as supervision for the detector's backbone.

The key idea is illustrated in Figure 1. The detector consists of a backbone ($\mathcal{B}$) and a head ($\mathcal{H}$), both of which receive the unmodified low-light input image ($x$). In parallel, the auxiliary pipeline processes $x$ through an enhancing module ($\mathcal{E}$) and a denoising module ($\mathcal{D}$), whose outputs are combined via Fourier-based fusion. This produces a supervision signal that guides the backbone via multi-task learning. After training, the auxiliary pipeline is removed, resulting in no inference overhead.

### 3.2 Auxiliary Modules for Self-Guided Supervision

To guide the backbone network toward improved representations in low-light settings, we design a target estimation pipeline that operates solely on the input image and is applied only during training. This pipeline consists of three self-supervised modules: an enhancing module, a denoising module, and a Fourier-based image fusion. The design explicitly addresses common low-light challenges such as insufficient brightness and high noise. All modules are retrained on the target dataset to best capture the characteristics of the given low-light domain.

**Enhancing Module ($\mathcal{E}$)**  To improve the overall brightness and contrast of the image, we incorporate a self-supervised low-light image enhancement network. This allows our framework to be trained without the need for paired ground-truth bright images, making it applicable to real-world scenarios where only low-light images are available. Low-light images inherently suffer from lim-

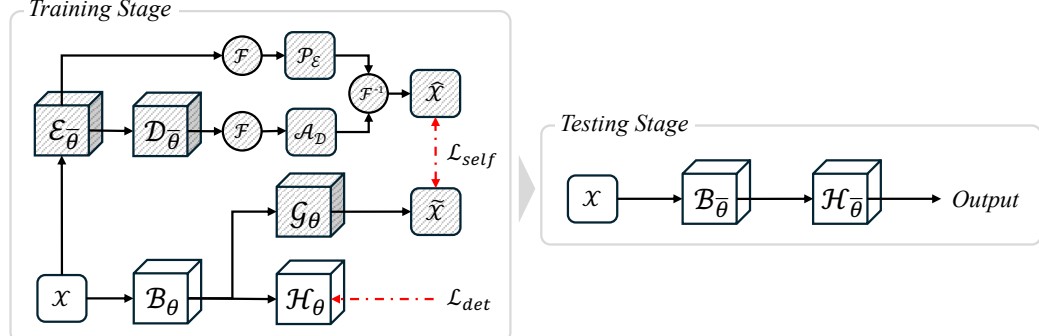

Figure 1: *Overview of the proposed framework.* We propose an auxiliary pipeline, depicted by dashed-line patterns, to generate a target image for self-guided supervision during training. This pipeline consists of a decoder ($\mathcal{G}$), an enhancing module ($\mathcal{E}$), a denoising module ($\mathcal{D}$), and Fourier-based image fusion ($\mathcal{F}, \mathcal{F}^{-1}$). At the testing stage, the auxiliary pipeline is removed, maintaining the original inference cost of the detector while improving its performance in low-light environments.

ited photon counts, resulting in a low Signal-to-Noise Ratio (SNR). While enhancement increases the signal strength of the image, it also amplifies noise, as illustrated in Fig. 2.

**Denoising Module ($\mathcal{D}$)**  To address noise-related challenges, we adopt a self-supervised denoising module. Since both image noise and perceptual details such as edges reside in the high-frequency domain, the denoising process reduces noise levels but also risks suppressing structural details—resulting in blurred or smoothed edges, as illustrated in Fig. 2.

**Fourier Transform ($\mathcal{F}$)**  The Fourier transform $\mathcal{F}$ operates on a given single-channel image $x$ of size $H \times W$, converting it into its complex component $X$ in the Fourier space.

$$X(u,v) = \sum_{h=0}^{H-1} \sum_{w=0}^{W-1} x(h,w) e^{-j2\pi(\frac{h}{H}u + \frac{w}{W}v)} \tag{1}$$

and inverse Fourier transform, denoted as $\mathcal{F}^{-1}$, reconstruct the original spatial domain image from its components of the frequency domain. In our pipeline, we apply the Fast Fourier Transform (FFT) (Nussbaumer & Nussbaumer, 1982) to each channel of an image independently, enabling a decomposition of frequency components across different spectral bands. As a result of the Fourier transform, the complex component $X(u,v)$ of an image can be decomposed into its amplitude $\mathcal{A}(X(u,v))$ and phase $\mathcal{P}(X(u,v))$.

$$\mathcal{A}(u,v) = \sqrt{R(X(u,v))^2 + I(X(u,v))^2}, \tag{2}$$

$$\mathcal{P}(u,v) = \tan^{-1}\left(\frac{I(X(u,v))}{R(X(u,v))}\right) \tag{3}$$

Here, $R(x)$ and $I(x)$ represent the real and imaginary parts of a complex value $X(u,v)$, respectively.

## 3.3 TARGET IMAGE CONSTRUCTION VIA FOURIER FUSION

According to Fourier theory, the amplitude component $\mathcal{A}$ captures low-level information such as brightness, noise, and style, while the phase component $\mathcal{P}$ preserves the structural and semantic details of an image (Piotrowski & Campbell, 1982; Xu et al., 2021). To retain the structural integrity of the enhanced image while incorporating the stylistic properties of the post-enhancement denoised image, we extract style from the denoised-after-enhancement result because the post-enhancement denoised image offers an illumination-corrected and noise-suppressed amplitude, providing a cleaner and more consistent source of stylistic information for fusion. (Fig. 2) FFT is applied independently to each RGB channel, with amplitude and phase computed separately per channel. We then combine the phase of the enhanced image with the amplitude of the denoised image and apply the inverse FFT (iFFT) to construct the final target image.

Specifically, the final supervisory target image $\hat{x}$ is estimated solely from the low-light input $x$ via the auxiliary pipeline: enhancement $x^{\mathcal{E}} = \mathcal{E}(x)$, denoising on enhanced image $x^{\mathcal{E}+\mathcal{D}} = \mathcal{D}(x^{\mathcal{E}})$, followed by Fourier decomposition. We extract the phase of the enhanced image $\mathcal{P}(x^{\mathcal{E}})$ and the amplitude of the denoised image $\mathcal{A}(x^{\mathcal{E}+\mathcal{D}})$, then reconstruct the fused target via inverse FFT:

$$\hat{x} = iFFT(\mathcal{A}(x^{\mathcal{E}+\mathcal{D}}) \cdot e^{j\mathcal{P}(x^{\mathcal{E}})}) \tag{4}$$

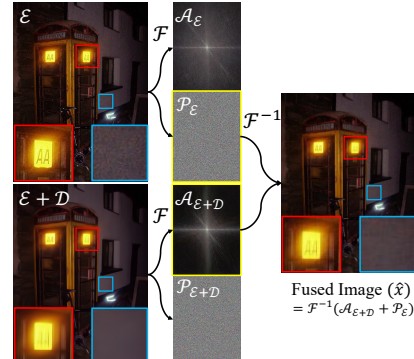

Figure 2: Fourier-based Image Fusion.

The fused image $\hat{x}$ serves as a dense, structure-preserving supervision that complements sparse detection labels and helps the backbone learn robust features under low-light and noisy conditions. Our framework tackles two key obstacles to using enhancement and denoising in detection: the auxiliary pipeline avoids inference-time cost, and Fourier fusion helps preserve boundaries and semantics.

## 3.4 FRAMEWORK INTEGRATION AND LOSS

Our self-guided detection framework is designed for seamless integration into standard object detectors without requiring any architectural modifications. During training, we attach the proposed detachable auxiliary decoder ($\mathcal{G}$) to the detector's backbone via skip connections, enabling multi-task learning. Each decoder layer receives feature maps from the corresponding backbone layer, facilitating dense supervisory signal with precise spatial alignment.

This dense supervision is provided by the fused target image $\hat{x}$, generated solely from the input low-light image using our Fourier Fusion module, as described in Section 3.3. Since the features from detector's backbone are used to predict this target image through the auxiliary decoder $\mathcal{G}$, the self-guided reconstruction loss ($\mathcal{L}_{self}$) encourages the backbone to retain structural and semantic cues that are often diminished in low-light conditions. Moreover, given that standard object detection targets (i.e., bounding boxes) provide sparse supervision, the self-guided reconstruction loss serves as a dense complementary signal. By enriching the overall supervision during training, the proposed framework enables the detector's backbone to learn more robust representations for object detection in challenging low-light environments.

**Self-Guided Supervision Loss** For the self-guided loss, which provides dense pixel-wise self-guidance to the detector's backbone, we use the mean squared error (MSE) between the decoder output $\tilde{x}$ and the fused target image $\hat{x}$:

$$\mathcal{L}_{self} = \|\tilde{x} - \hat{x}\|_2^2 \tag{5}$$

The total training loss is thus defined as:

$$\mathcal{L}_{total} = \mathcal{L}_{det} + \lambda \cdot \mathcal{L}_{self} \tag{6}$$

where $\lambda$ is a weighting factor that balances detection and self-guidance.

## 4 EXPERIMENTS

In this section, we evaluate our framework on three public datasets with multiple detectors to demonstrate its effectiveness and generalization capability. We conduct experiments on face detection (Sec. 4.2), multi-class object detection using YOLOv3 (Redmon, 2018) (Sec. 4.3), and nighttime object detection for autonomous driving (Sec. 4.4). We further provide an in-depth analysis (Sec. 4.5) and ablation studies (Sec. 4.6).

### 4.1 IMPLEMENTATION DETAILS

Our model is trained using the proposed Self-Guided Low-light Detection (SGLDet) framework, with SCI (Ma et al., 2022a) as the enhancing module ($\mathcal{E}$) and SDAP (Pan et al., 2023) as the de-

noising module ($\mathcal{D}$). Since both modules are able to be trained in a self-supervised manner, they are retrained on the target dataset. For the auxiliary detachable decoder ($\mathcal{G}$), we employ a U-Net decoder architecture composed of transposed convolutions and double convolution blocks. To directly influence all layers of the detector's backbone, each backbone layer is connected to the decoder via skip connections, which are used only during training.

For benchmarking, we compare our framework against state-of-the-art LLIE methods (Liu et al., 2021a; Guo et al., 2020; Li et al., 2021; Ma et al., 2022a; Cai et al., 2023; Zhou et al., 2024) and T2 (Cui et al., 2024), a dedicated low-light detector, that we reimplemented for evaluation. To ensure a fair comparison, self-supervised LLIE methods, which can be trained from unpaired low-light images, were retrained on the target dataset. Meanwhile, supervised methods were trained on the LOL dataset (Chen Wei, 2018), which provides paired ground truth normal images.

## 4.2 EVALUATIONS ON THE DARK FACE DATASET

In this experiment, we use DARK FACE (Yang et al., 2020) dataset, designed for low-light face detection. Since the labeled faces are small in nighttime scens, this dataset is well-suited for evaluating detection models under challenging conditions.

**Settings** For the baseline detector, we employ DSFD (Li et al., 2019) with ResNet (He et al., 2016) backbone. Since our objective is to evaluate detection performance in low-light conditions, all models are trained from scratch without using any pre-trained weights, except for initializing the ResNet backbone with its default weights. The DARK FACE dataset consists of 6,000 real-world low-light images, following the HLA-Face (Wang et al., 2021) split, where 5,500 images are used for training and 500 for validation. We optimize the model using SGD, with the first 10% of 100 epochs allocated for warming up. The maximum learning rate is set to 0.001, and experiments are conducted using a cosine one-cycle scheduler. In our proposed approach, $\mathcal{L}_{self}$ is formulated with MSE, and its weight is scaled by a factor of 0.01. The computational cost of each method was measured on an NVIDIA H100 GPU. For fair comparison, supervised LLIE methods were trained with paired normal-light images from the LOLv1 dataset, and the domain adaptation method DAI-Net was additionally pre-trained on WIDER Face (Yang et al., 2016) as the source-domain dataset and trained on the DARK FACE, following the protocol outlined by the original authors.

**Results** As shown in Table 1, our framework achieves the highest performance gain (+16.9 mAP) with no additional parameters, FLOPs, or inference time, highlighting its practical effectiveness. Interestingly, supervised LLIE methods trained on LOLv1 (68.1 and 65.9 in mAP) are higher than self-supervised approaches retrained on DARK FACE (from 61.1 to 65.7 in mAP), suggesting their better adaptation to the dataset's low-light conditions. DAI-Net, a domain adaptation method pre-trained on WIDER Face, also demonstrates competitive performance (68.5 in mAP) without inference-time overhead. However, our method ultimately outperforms all competitors (Table 1, Figure 3). By improving backbone features without extra modules or datasets, it achieves state-of-the-art accuracy.

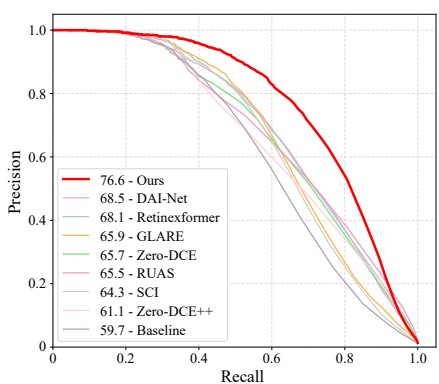

Figure 3: PR curve on the DARK FACE.

As shown in Fig. 4, our method reduces missed detections—especially for small, distant faces—by enhancing night-specific feature learning in low-light and noisy conditions. The results confirm the effectiveness of our framework for the detection in real-world scenes.

## 4.3 EVALUATIONS ON THE EXDARK DATASET

The ExDark dataset (Loh & Chan, 2019) is a multi-class object detection benchmark widely adopted in the low-light detection research. Following the protocol established by MAET (Cui et al., 2021), we adopt YOLOv3 as the baseline detector, which serves as a standard evaluation setup for this task.

Table 1: Performance and computational cost comparison with state-of-the-art on DARK FACE.

| Category | Method | mAP | | Additional Dataset | Additional Costs Params | FLOPs | TIME (ms) |
|---|---|---|---|---|---|---|---|
| Baseline Detector | DSFD (Li et al., 2019) | 59.7 | - | - | - | - | - |
| + Self Supervised LLIE | RUAS [CVPR 2021] | 65.5 | (+5.8) | | 3.4 K | 5.23 G | 4.98 |
| | Zero-DCE [CVPR 2020] | 65.7 | (+6.0) | | 79.4 K | 123.21 G | 4.30 |
| | Zero-DCE++ [TPAMI 2021] | 61.1 | (+1.4) | - | 10.6 K | 0.15 G | 1.05 |
| | SCI [CVPR 2022] | 64.3 | (+4.6) | | 0.3 K | 0.38 G | 0.71 |
| + Supervised LLIE | Retinexformer [ICCV 2023] | 68.1 | (+8.4) | LOLv1 | 1.61 M | 385.88 G | 43.73 |
| | GLARE [ECCV 2024] | 65.9 | (+6.2) | | 71.72 M | 74.26 T | 746.72 |
| + Domain Adaptation | DAI-Net [CVPR 2024] | 68.5 | (+8.8) | WIDER Face | - | - | - |
| + Self-Guided framework | **Ours** | **76.6** | **(+16.9)** | **-** | **-** | **-** | **-** |

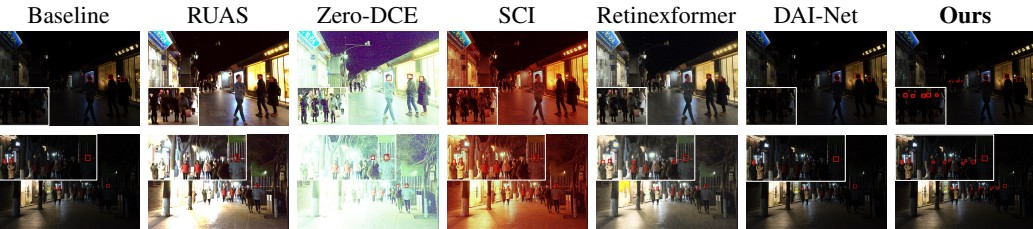

Baseline    RUAS    Zero-DCE    SCI    Retinexformer    DAI-Net    **Ours**

Figure 4: Qualitative comparisons on DARK FACE. More results can be found in the Supplementary.

By benchmarking our method on this representative setting, we aim to demonstrate the effectiveness and generalizability of our framework in comparison to state-of-the-art approaches.

**Settings** We use YOLOv3-608 (Redmon, 2018) with Darknet-53 backbone as our baseline detector. The ExDark (Loh & Chan, 2019) dataset consists of 7,363 images. Following the MAET (Cui et al., 2021) protocol, we use 5,896 images for training and 1,467 for validation, strictly adhering to their official split and training pipeline. Except for domain adaptation methods, all models including ours are initialized with COCO-pretrained weights for the YOLOv3 detector and finetuned on Ex-Dark for 25 epochs, as done in the MAET comparisons. For domain adaptation models, we follow their respective pretraining strategies on COCO and perform 25-epoch finetuning on ExDark. We optimize all models using SGD, with 2,000 warm-up iterations and a learning rate of 0.001, reduced by a factor of 10 at epochs 18 and 23, following the training schedule of MAET.

**Results** Table 2 presents the quantitative results of our framework compared to existing methods on the ExDark (Loh & Chan, 2019) dataset. Several self-supervised LLIE methods such as RUAS (Liu et al., 2021a) and Zero-DCE++ (Li et al., 2021) degrade performance (-2.7 and -1.1 mAP, respectively). In contrast, SCI (Ma et al., 2022a) and Zero-DCE (Guo et al., 2020) show modest gains (+1.5 and +0.5 mAP), suggesting that lightweight enhancement can help but is not consistently reliable.

Supervised LLIE methods like Retinexformer (Cai et al., 2023) and GLARE (Zhou et al., 2024), despite leveraging paired training data, fail to outperform the baseline (-0.9 and -1.4 mAP). Domain adaptation methods, MAET (Cui et al., 2021) and DAI-Net (Du et al., 2024), perform better (+1.3 and +1.9 mAP), benefiting from cross-domain training. Our framework achieves the highest gain of +2.2 mAP over the baseline, establishing a new state-of-the-art result on ExDark.

### 4.4 NIGHT OBJECT DETECTION FOR DRIVING SCENES

Object detection in nighttime driving scenarios is particularly challenging due to extremely low illumination and severe sensor noise. Unlike general low-light photography, where exposure can be adjusted, autonomous driving systems use fixed camera settings for day–night consistency. This results in nighttime images with severely degraded visibility and object boundaries. Moreover, the nighttime scenes in nuImages frequently exhibit intense headlight glare, high-frequency noise along

Table 2: Quantitative results on the ExDark.

| Category | Method | mAP |
|---|---|---|
| Baseline Detector | YOLOv3-608 | 76.4 |
| + Retinex Decomposition | T2 [AAAI 2024] | 75.5 |
| + Self Supervised LLIE | RUAS [CVPR 2021]
Zero-DCE [CVPR 2020]
Zero-DCE++ [TPAMI 2021]
SCI [CVPR 2022] | 73.7
76.9
75.3
77.9 |
| + Supervised LLIE | Retinexformer [ICCV 2023]
GLARE [ECCV 2024] | 75.5
75.0 |
| + Domain Adaptation | MAET [ICCV 2021]
DAI-Net [CVPR 2024] | 77.7
78.3 |
| + Self-Guided framework | **Ours** | **78.6** |

Table 3: Experiments on nuImages.

| Model | mAP | Total Inference Time | |
|---|---|---|---|
| | | TIME (ms) ↓ | FPS ↑ |
| *Training from scratch on Night (w/o pretraining)* | | | |
| YOLOv8-m | 52.4 | **5.88** | **170.1** |
| + T2 | 51.1 | 12.56 | 79.6 |
| + SCI | 53.5 | 8.12 | 123.2 |
| + Retinexformer | 52.1 | 129.24 | 7.3 |
| + GLARE | 51.7 | 4896.72 | 0.2 |
| **YOLOv8-m + Ours** | **55.3** | **5.88** | **170.1** |
| *Pretrained on Daytime → Finetuned on Night (w/ pretraining)* | | | |
| YOLOv8-m | 56.4 | | |
| + MAET | 53.8 | **5.88** | **170.1** |
| + DAI-Net | 54.4 | | |
| **YOLOv8-m + Ours** | **58.0** | **5.88** | **170.1** |

road boundaries, and pronounced illumination imbalance between foreground objects and the background, further complicating reliable feature extraction.

To assess our framework's performance in these conditions, we use the nuImages (Caesar et al., 2020) dataset, chosen for its authentic nighttime driving images. We measure both detection accuracy and inference speed to verify its suitability for real-time driving applications.

**Settings** To evaluate the practicality of our method, we adopt YOLOv8-m (Jocher et al., 2023), a modern real-time object detector, as the baseline. The nuImages dataset primarily consists of daytime scenes, with only a limited number of nighttime images. From this dataset, we extract 4,485 nighttime images and split[1] them into 3,591 for training and 894 for validation. Due to severe class imbalance—e.g., 6,150 instances of *car* but only 66 of *bicycle*—we focus on four categories with sufficient samples: car (6,150), pedestrian (1,665), traffic cone (1,962), and barrier (2,202).

We consider two training settings: (i) training from scratch using only nighttime images, and (ii) finetuning on nighttime images after pretraining on daytime nuImages. The second setting is crucial for comparing with domain adaptation methods like MAET and DAI-Net, as they require a source (daytime) domain for pretraining. Our framework and the baseline were tested in both settings, while the domain adaptation methods were evaluated only in the second, inherently. All models are trained under identical settings, and inference speed is measured on an NVIDIA H100 GPU.

**Results** Table 3 presents the results on nighttime driving scenes from the nuImages dataset. Our framework achieves the best performance in both training scenarios while maintaining real-time inference speed. When trained from scratch on nighttime images, our method surpasses the YOLOv8-m baseline by +2.9 mAP with no additional inference cost. Notably, most enhancement or decomposition methods fail to improve performance (e.g., -1.3 with T2, -0.7 with GLARE) and lose real-time capability (e.g., 0.2 FPS with GLARE). In contrast, our method enhances

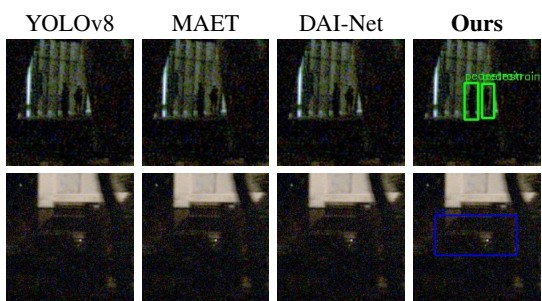

YOLOv8    MAET    DAI-Net    **Ours**

Figure 5: Qualitative comparison on nuImages with domain adaptation.

detection accuracy without compromising speed (170.1 FPS), highlighting its efficiency for real-world deployment. In the domain adaptation setting (pretraining on daytime data, finetuning on night images), our framework again delivers the highest accuracy, outperforming both the baseline (+2.6 mAP) and other methods like MAET (-2.6 mAP) and DAI-Net (-2.0 mAP). These results highlight a critical limitation of these techniques for driving scenarios with large domain gaps: they often fail to adapt from a source domain (daytime) to a vastly different target domain (nighttime). Overall, our approach demonstrates state-of-the-art performance on this challenging real-world benchmark, achieving robust generalization without sacrificing computational efficiency.

---

[1]Since the original nuImages validation set does not include night scenes, we resplit the dataset.

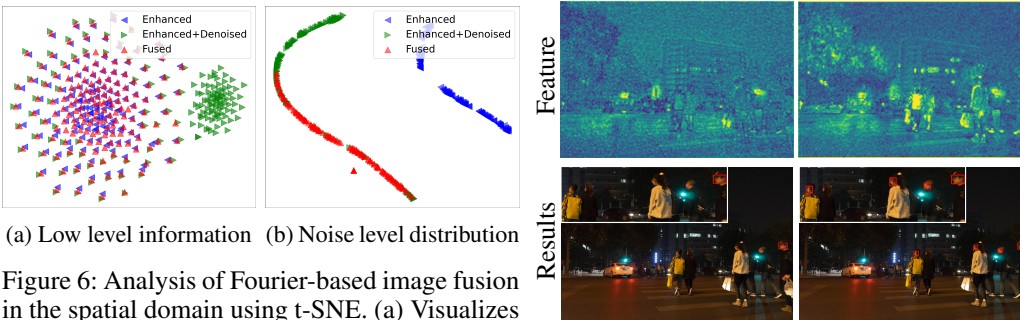

(a) Low level information  (b) Noise level distribution

Figure 6: Analysis of Fourier-based image fusion in the spatial domain using t-SNE. (a) Visualizes low-level information, including structural and semantic details, while (b) illustrates the distribution of noise levels.

Figure 7: Feature visualization.

## 4.5 ANALYSIS OF THE SGLDET FRAMEWORK

**Target Image Analysis**   To verify whether the fused target image in Sec 3.3 retains low-level spatial information while exhibiting a noise level similar to that of the denoised image, we perform an analysis using t-SNE (Van der Maaten & Hinton, 2008). We extract low-level information using HOG features (Dalal & Triggs, 2005) and assessed noise levels by subtracting a median-filtered image. The results indicate that the fused target image retains the low-level traits of the enhanced image (6-a) while its noise level matches that of the denoised image (6-b).

**Backbone Feature Analysis**   Since the proposed framework incorporates an auxiliary task connected to the backbone during training, it enhances feature representation in low-light conditions. Fig. 7 visualizes feature maps from the same ResNet backbone for both the baseline and our method. With only sparse box-level supervision, the baseline exhibits weak activations, often missing clear object-background separation. In contrast, our self-guided framework yields stronger activations in the person region and sharper boundary contrast, indicating improved localization and structure awareness[2]. These results confirm that the fused pixel-level guidance helps the backbone learn more discriminative and robust features, ultimately improving detection performance.

**Sensitivity to the Weighting Hyperparameter** $\lambda$   We further analyze the sensitivity of the framework to the weighting factor $\lambda$ between $L_{\text{det}}$ and $L_{\text{self}}$ on the DARK FACE dataset. Small values of $\lambda$ (e.g., $\leq 10^{-3}$) provide only limited dense guidance, resulting in moderate gains. Performance peaks at $\lambda = 0.01$, indicating that a balanced contribution from reconstruction supervision most effectively complements the detection objective. In contrast, large values ($\lambda \geq 0.05$) cause the reconstruction loss to dominate, degrading detection features and leading to unstable optimization when $\lambda$ becomes too large. These results suggest that $\lambda$ should remain within a moderate range for stable and effective training.

Table 4: Sensitivity analysis.

| $\lambda$ | mAP |
|---|---|
| 0.0001 | 72.5 |
| 0.0010 | 74.0 |
| 0.0100 | **76.6** |
| 0.0500 | 67.3 |
| 0.1000 | 61.6 |

## 4.6 ABLATION STUDIES

**Ablation on Framework Components**   Table 5 presents an ablation study on target image guidance using DARK FACE dataset. First, when using enhancing-only image (i.e., processed solely through the Enhancing Module), brightness and contrast are improved, resulting in a +13.2 mAP increase over the baseline. However, as shown in Fig. 2, this approach also amplifies noise. To mitigate this, applying a Denoising Module

Table 5: Ablation on framework components.

| Method | $\mathcal{E}$ | $\mathcal{D}$ | Fusion $(\mathcal{E}, \mathcal{D})$ | mAP |
|---|---|---|---|---|
| DSFD | | | | 59.7 |
| | ✓ | | | 72.9 |
| | ✓ | ✓ | | 72.2 |
| Ours | ✓ | ✓ | ✓ | **76.6** |

after enhancement effectively suppresses noise, but at the cost of some boundary detail loss, leading to a slight performance drop to 72.2 mAP. Finally, leveraging Fourier-based Image Fusion, which combines both results in the Fourier domain, enables the target image to retain structural details

---

[2]Further analysis on improved localization is provided in the Supplementary Material.

while effectively suppressing noise, achieving 76.6 mAP. This represents an +16.9 mAP improvement over the baseline, validating the effectiveness of each component.

**Ablation on Module Choices** In addition to the component-wise analysis, we investigate how sensitive our framework is to the specific choices of the Enhancer ($\mathcal{E}$) and Denoiser ($\mathcal{D}$) using various combinations in Table 6. Even a naive setting with gamma correction and Gaussian blur already yields a large improvement over the baseline (59.7 mAP $\rightarrow$ 70.9 mAP), indicating that the gain does not rely on sophisticated modules. When employing the denoiser to SDAP, self-supervised enhancers such as Zero-DCE and SCI, which can be trained directly on the target dataset, achieve higher performance than the supervised Retinexformer that requires paired illumination labels, suggesting that our self-guided pathway is naturally aligned with target-adaptable enhancement. Conversely, applying the enhancer to SCI and varying the denoiser shows that Gaussian-trained Restormer (Zamir et al., 2022) variants are highly sensitive to the assumed noise level $\sigma$ and can even underperform a simple median blur when the noise model is mismatched, whereas Restormer trained on real noise dataset (Abdelhamed et al., 2018) and SDAP provide more stable improvements because both rely on more realistic noise assumptions. SDAP achieves the best mAP through self-supervised training on the target domain. Overall, these trends demonstrate that while stronger enhancement or denoising modules can lead to incremental accuracy gains, the primary performance improvement consistently originates from the detachable self-guided pathway itself, indicating that our framework remains robust and flexible with respect to module choice.

Table 6: Ablation study on module choices.

| Enhancer ($\mathcal{E}$) | Denoiser ($\mathcal{D}$) | mAP |
|---|---|---|
| Gamma correction | Gaussian blur | 70.9 |
| Retinexformer | | 72.5 |
| Zero-DCE | SDAP | 73.8 |
| SCI | | **76.6** |
| | Median blur | 73.6 |
| | Restormer ($\sigma = 15$) | 75.0 |
| SCI | Restormer ($\sigma = 25$) | 73.0 |
| | Restormer ($\sigma = 50$) | 73.0 |
| | Restormer (real-noise) | 76.0 |
| | SDAP | **76.6** |

## 5 CONCLUSION

In this paper, we proposed a self-guided framework for enhancing object detection performance in low-light environments. Our approach incorporates auxiliary supervision components during training while maintaining the original computational cost at inference time, distinguishing it from methods that require additional inference modules or depend on domain adaptation strategies. The framework integrates self-supervised modules for enhancement, denoising, and Fourier-domain fusion to improve backbone feature representations under challenging lighting conditions. Experiments across three standard low-light benchmarks (DARK FACE, ExDark, and nuImages) demonstrate that our framework consistently outperforms existing approaches, achieving new state-of-the-art results without modification to the original detection architecture at inference. On the nuImages dataset, which contains nighttime driving scenes, our framework outperforms both enhancement-based methods that incur runtime overhead and domain adaptation approaches in scenarios with large domain gaps. These results demonstrate that our training-time supervision strategy provides an effective solution for low-light object detection in practical applications.

## ACKNOWLEDGMENTS

This work was supported by the National Research Foundation of Korea (NRF) grant funded by the Korea government (MSIT) (No. RS-2024-00409492).

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

## LLM USAGE DISCLOSURE

We used a large language model (LLM) solely to aid in polishing the writing of this paper. All research ideas, methods, experiments, and analyses are the sole work of the authors.

