## A    EFFECT OF DENSE SUPERVISION ON LOCALIZATION ACCURACY

To analyze the effect of our self-guided framework on localization, we evaluate detection performance across multiple IoU thresholds, ranging from 0.5 to 0.95. As illustrated in Figure 1, the performance gap between our method and the baseline increases consistently as the IoU threshold becomes more strict. This trend suggests that our framework not only improves overall detection robustness under low-light conditions but also leads to more precise object localization.

We believe this improvement is largely due to the dense pixel-level supervision introduced during training through the fused target image. By complementing sparse bounding box annotations with structure-preserving guidance, the backbone learns more discriminative features that enable tighter and more accurate object boundaries. These results support our hypothesis that dense supervision encourages spatially coherent and localization-aware feature representations.

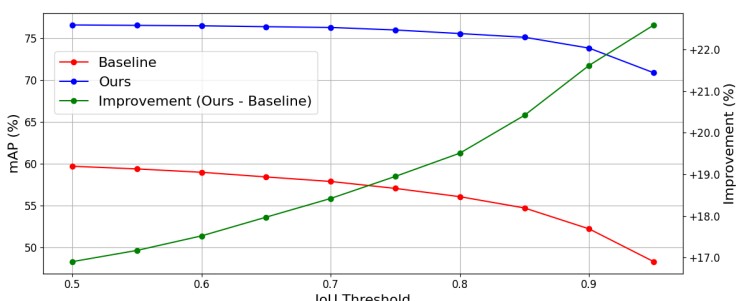

Figure 1: Detection mAP under different IoU thresholds on the DARK FACE dataset. As the IoU threshold increases from 0.5 to 0.95, the performance gap between our method and the baseline consistently widens, demonstrating improved localization accuracy enabled by our structure-aware supervision.

## B    PERFORMANCE VARIATION ACROSS LOW-LIGHT BENCHMARKS

Our framework consistently improves detection performance across all three low-light benchmarks: DARK FACE, ExDark, and nuImages. However, we observe that the magnitude of improvement varies across datasets, with the largest gain observed on DARK FACE. To better understand this discrepancy, we analyze the average relative brightness of each dataset, as shown in Figure 2. Among the three, DARK FACE contains the darkest scenes—nearly two times darker than ExDark and over five times darker than daytime nuImages. In such extreme low-light conditions, the pixel-level supervision provided by our framework becomes especially beneficial, as it effectively encourages the learning of structure-preserving features from low-contrast input image. In contrast, ExDark and nuImages (Night) contain relatively brighter images than DARK FACE, where standard detectors are already able to extract salient features. In these cases, while our framework still yields noticeable improvements, the relative gains are naturally smaller due to the less severe degradation in the input signal. This analysis highlights the strength of our approach under the most challenging illumination scenarios and explains the dataset-specific variation in performance improvements.

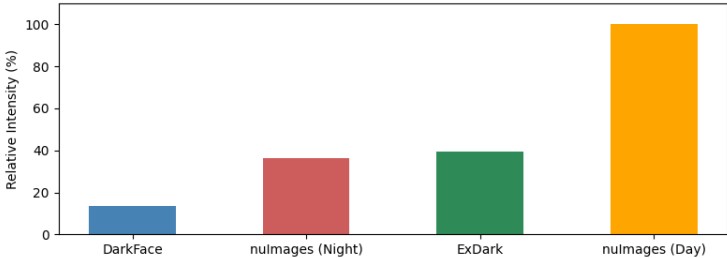

Figure 2: Relative brightness of each dataset, normalized to the mean intensity of nuImages (Day).

## C  ENSURING FAIR AND REPRODUCIBLE COMPARISON ON EXDARK

**Splits and Pretrained Weights.**  For all experiments on the ExDark dataset, we strictly followed the evaluation protocol and dataset split introduced by MAET (Cui et al., 2021). To ensure a fair and reproducible comparison, we adopted the official weights released by the authors of each method. Specifically, the baseline YOLOv3 model as well as KinD (Zhang et al., 2019), Zero-DCE (Guo et al., 2020), and MAET (Cui et al., 2021) models were evaluated using the official pre-trained weights provided in the MAET GitHub repository. We independently verified all results by running inference using these weights, ensuring reproducibility under the same experimental conditions. For DAI-Net (Du et al., 2024), we used the weights publicly released by its authors, who trained the model using the same protocol as MAET. These weights are compatible with the same dataset split and detector architecture, ensuring a fair comparison.

**Updated Results of MAET.**  Importantly, we note that the scores reported in the original MAET and DAI-Net papers for several baseline and enhancement methods (e.g., KinD, Zero-DCE) have since been revised by the MAET authors. The updated values, along with corresponding weights, are now available on the official MAET GitHub repository[1]. In this work, we adopt these corrected scores to provide the most up-to-date and consistent comparison across all methods. For more recent enhancement methods that were not included in MAET or DAI-Net (e.g. SCI, Retinexformer) we conducted the evaluations ourselves using their official codebases. Each method was trained and tested under exactly the same conditions as MAET, using the YOLOv3-608 detector, COCO pre-training, and the same dataset split and training schedule. This ensures that all results are directly comparable and evaluated under a unified protocol.

| Method | Bicycle | Boat | Bottle | Bus | Car | Cat | Chair | Cup | Dog | Motorbike | People | Table | mAP |
|---|---|---|---|---|---|---|---|---|---|---|---|---|---|
| Baseline (Redmon, 2018) | 79.8 | 75.3 | 78.1 | 92.3 | 83.0 | 68.0 | 69.0 | 79.0 | 78.0 | 77.3 | 81.5 | 55.5 | 76.4 |
| KinD (Zhang et al., 2019) | 80.1 | 77.7 | 77.2 | 93.8 | 83.9 | 66.9 | 68.7 | 77.4 | 79.3 | 75.3 | 80.9 | 53.8 | 76.3 |
| RUAS (Liu et al., 2021) | 80.4 | 75.9 | 74.8 | 89.8 | 79.3 | 69.8 | 67.4 | 70.2 | 74.6 | 70.4 | 78.0 | 54.4 | 73.7 |
| Zero-DCE (Guo et al., 2020) | **84.1** | 77.6 | 78.3 | 93.1 | 83.7 | 70.3 | 69.8 | 77.6 | 77.4 | 76.3 | 81.0 | 53.6 | 76.9 |
| Zero-DCE++ (Li et al., 2021) | 79.4 | 76.0 | 74.8 | 91.6 | 83.3 | 69.0 | 68.0 | 72.3 | 78.6 | 75.2 | 78.7 | 56.4 | 75.3 |
| SCI (Ma et al., 2022) | 82.0 | 77.9 | 78.1 | 92.9 | 83.9 | 72.9 | 71.8 | 78.0 | 81.0 | 75.9 | **82.7** | 57.2 | 77.9 |
| Retinexformer (Cai et al., 2023) | 80.9 | 72.8 | 75.6 | 91.0 | 83.4 | 71.7 | 69.7 | 72.1 | 76.5 | 76.9 | 80.6 | 54.7 | 75.5 |
| GLARE (Zhou et al., 2024) | 79.3 | 73.3 | 74.1 | 90.5 | 83.5 | 68.2 | 69.3 | 75.1 | 78.0 | 74.0 | 79.2 | 55.2 | 75.0 |
| T2 (Cui et al., 2024) | 82.1 | 76.5 | 77.4 | 89.6 | 82.8 | 67.1 | 67.8 | 75.3 | 77.4 | 74.7 | 80.3 | 54.5 | 75.5 |
| MAET (Cui et al., 2021) | 83.1 | 78.5 | 75.6 | 92.9 | 83.1 | 73.4 | 71.3 | 79.0 | 79.8 | 77.2 | 81.1 | 57.0 | 77.7 |
| DAI-Net (Du et al., 2024) | 83.8 | 75.8 | 75.1 | **94.2** | 84.1 | 74.9 | 73.1 | 79.2 | 82.2 | 76.4 | 80.7 | **59.8** | 78.3 |
| Ours | 82.0 | **79.9** | **80.6** | 93.6 | **85.1** | 71.2 | 71.7 | **79.2** | 80.0 | **79.8** | 82.4 | 57.8 | **78.6** |

Table 1: Per-class detection mAP (%) comparison on the ExDark dataset using YOLOv3. All models were evaluated under the MAET protocol. Baseline, KinD, Zero-DCE, and MAET results were reproduced using official weights provided by the MAET authors. DAI-Net results are based on the weights released by its authors under the same protocol. Note that some of the reported values (e.g., Baseline, KinD, Zero-DCE) were updated by the MAET authors after publication, and our evaluation reflects these revised results. Our method achieves the highest overall mAP among all compared approaches.

---

[1]MAET GitHub: `https://github.com/cuiziteng/ICCV_MAET`.

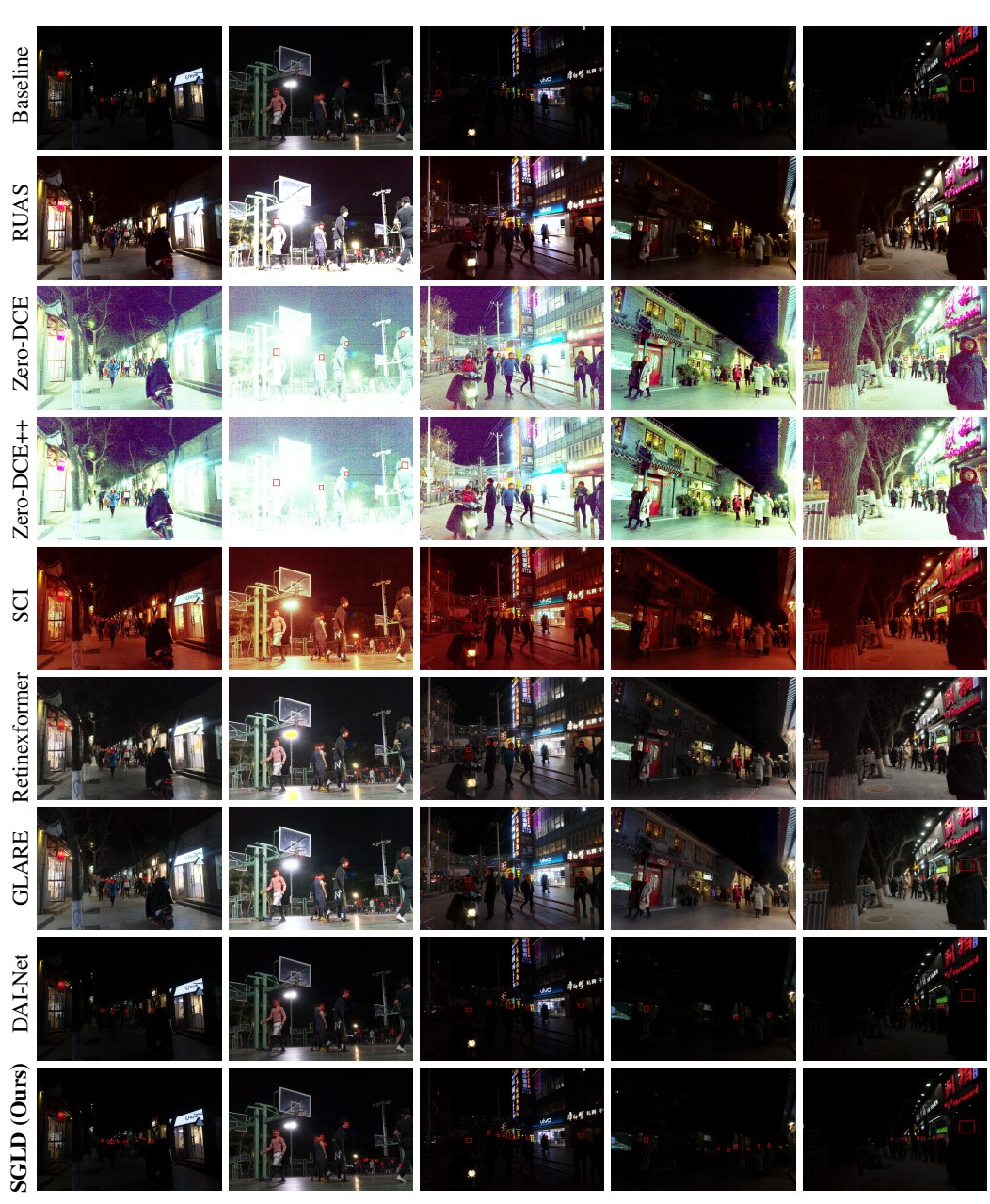

Figure 3: Additional Results on DARK FACE (Yang et al., 2020) dataset. *Zoom in for best view.*

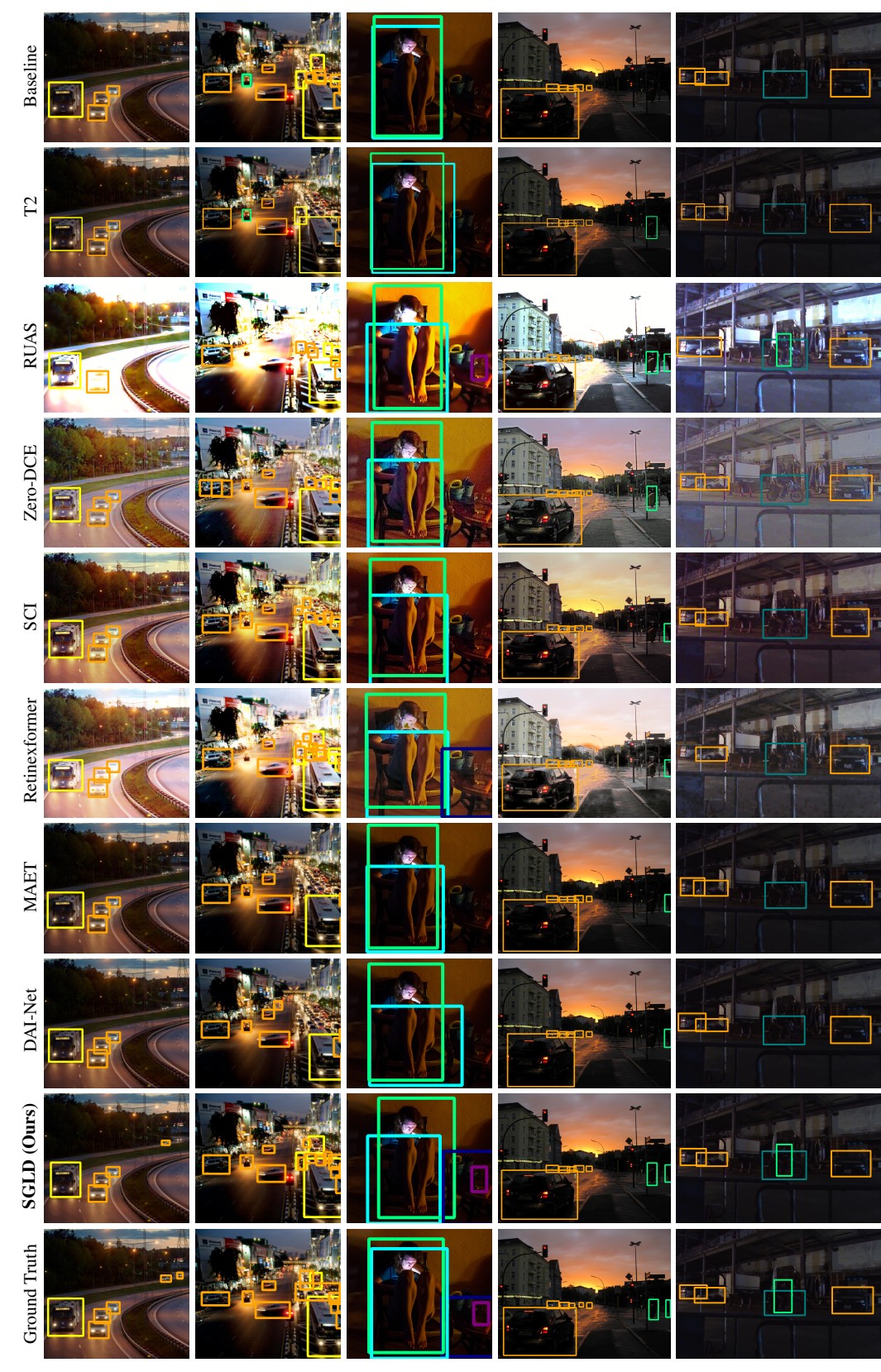

Figure 4: Additional Results on ExDark (Loh & Chan, 2019) dataset.

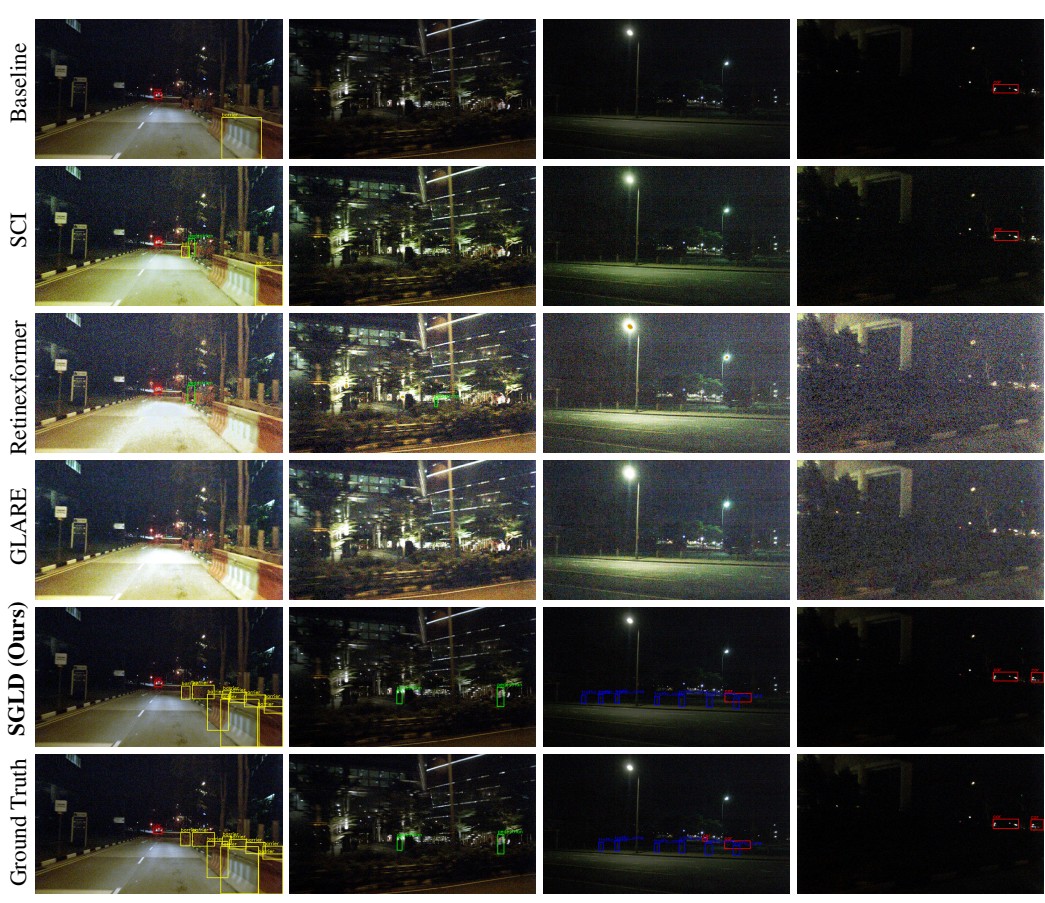

Figure 5: Additional qualitative comparison with enhancement-based method on nuImages (Caesar et al., 2020).

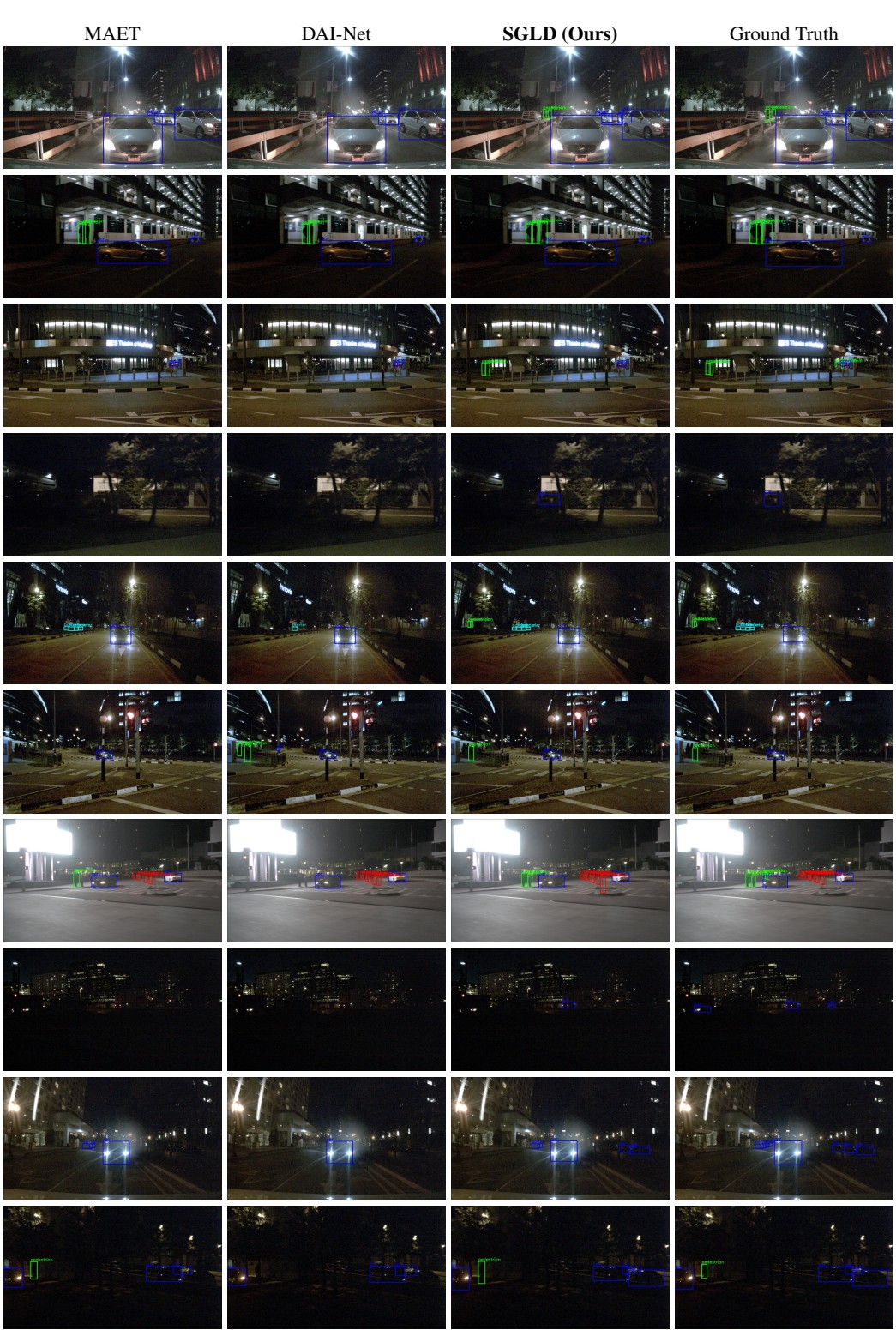

Figure 6: Additional qualitative comparison with domain adaptation method on nuImages (Caesar et al., 2020).