# OpenReview forum: "Self-Guided Low Light Object Detection Framework"
_ICLR.cc/2026/Conference — ICLR 2026 Poster_

### Official Review · Reviewer_rmxc · 2025-10-30

**Soundness:** 4
**Presentation:** 3
**Contribution:** 3
**Rating:** 6
**Confidence:** 3

**Summary:**

This paper proposes a Self-Guided Low-light Object Detection Framework with a detachable auxiliary pipeline used only during training, which generates a high-quality supervisory target through an image enhancement module, a denoising module, and Fourier-domain fusion. Experimental results show that this method achieves state-of-the-art performance on multiple benchmarks, with a significant improvement on DARK FACE.

**Strengths:**

1. The designed auxiliary pipeline is activated only during training and is completely detached at inference time, which allows the model to achieve substantial performance gains while maintaining the exact same inference speed.
2. The method leverages the properties of the Fourier transform and self-supervised training strategy, which is novel and sound.
3. The writing is well-organized and easy to follow.
4. Experiments show significant improvement on multiple benchmarks.

**Weaknesses:**

1. The paper repeatedly emphasizes "zero inference overhead", but completely omits any discussion of the cost during training-stage. The introduction of the auxiliary pipeline ($\mathcal{E}, \mathcal{D}, \mathcal{G}$), retraining of $\mathcal{E}$ and $\mathcal{D}$ on the target dataset, and multiple Fourier transforms will clearly increase training complexity and time, but this trade-off is not quantified.
2. The total loss function uses a hyperparameter $\lambda$ to balance the main and auxiliary pipeline, but the paper lacks a sensitivity analysis for this crucial hyperparameter.

**Questions:**

In Section 3.3, the authors chose the serial strategy $x^{\mathcal{E}+\mathcal{D}} = \mathcal{D}(x^{\mathcal{E}})$. Why is the current serial strategy the superior choice? Why not choose a parallel fusion strategy (e.g., fusing $\mathcal{P}(\mathcal{E}(x))$ and $\mathcal{A}(\mathcal{D}(x))$, where $\mathcal{D}$ operates on the original input $x$)?

---

> ### Author Response · Authors · 2025-11-21
> **Response to Reviewer rmxc (1/3)**
>
> We thank Reviewer rmxc for acknowledging the soundness and contributions of our framework and for the positive assessment. We also appreciate the constructive feedback on training-time cost, hyperparameter sensitivity, and our choice of the serial fusion strategy. These comments helped us strengthen the clarity and justification of our design. Below, we address each point in detail with supporting quantitative analyses.
>
> ### **1. Discussion of the cost during training**
>
> As the reviewer correctly pointed out, our paper repeatedly emphasizes that the proposed framework introduces zero additional computation during inference, while the training stage necessarily involves extra cost due to the auxiliary pipeline.
>
> Under the DARK FACE + DSFD setting, using identical hardware and batch size, the measured per-epoch training time is as follows:
>
> | Model | Training Time per Epoch | Relative Cost |
> | --- | --- | --- |
> | Baseline (DSFD) | 2 min 33 sec | 1.0x |
> | Ours (Self-Guided) | 4 min 39 sec | ~1.8x |
>
> Thus, our self-guided framework requires approximately 1.8x longer training time per epoch. This increase primarily comes from the auxiliary pipeline, including the U-Net-based decoder. However, this overhead is paid only once during training, and the trained detector is used with no additional cost at deployment. In practical scenarios such as DARK FACE or nuImages, where the emphasis is on real-time performance in real-world applications, a one-time 1.8x increase in training time is far more acceptable than methods that increase inference-time overhead. For example, LLIE-based preprocessing methods require running enhancement models on every input image at deployment, and domain adaptation approaches often rely on large external source datasets (e.g., WIDER Face or daytime COCO) together with extensive additional pre-training.
>
> In contrast, our framework avoids these issues: It does not require external datasets or additional large-scale pretraining, does not modify the detector architecture, and keeps the inference pipeline entirely unchanged. The detector therefore runs at the same speed as the baseline model while benefiting from stronger low-light representations acquired during training.
>
> Overall, although our approach introduces some training-time overhead, it is a deliberate and practical design choice that avoids any additional inference-time cost while achieving substantial performance improvements.

---

> ### Author Response · Authors · 2025-11-21
> **Response to Reviewer rmxc (2/3)**
>
> ### **2. Sensitivity analysis of hyperparameter** $\lambda$
>
> We thank both Reviewer rmxc and Reviewer Y1L8 for raising the importance of analyzing the weighting factor between the detection loss $L_{\text{det}}$ and the self-guided reconstruction loss $L_{\text{self}}$. Since our framework jointly optimizes sparse detection supervision and dense pixel-level guidance, understanding the sensitivity to the hyperparameter $\lambda$ is indeed crucial. In response, we conducted an extensive study across a wide range of $\lambda$ values, and the results are summarized in the table below.
>
> | Hyperparameter $\lambda$ | AP |
> | --- | --- |
> | 0.0001 | 72.5 |
> | 0.0010 | 74.0 |
> | 0.0100 | **76.6** |
> | 0.0500 | 67.3 |
> | 0.1000 | 61.6 |
> | 0.5000 | Did not converge |
> | 1.0000 | Did not converge |
>
> Several trends can be observed from the results:
>
> **(1) Small $\lambda$: insufficient dense supervision.**
>
> When $\lambda$ is too small, the influence of $L_{\text{self}}$ becomes marginal. Although the backbone still benefits from a small amount of dense guidance, the improvement remains limited, as observed at $\lambda \le 10^{-3}$.
>
> **(2) Moderate $\lambda$: optimal balance.**
>
> We observe performance improvements as $\lambda$ increases from $10^{-4}$ to $10^{-2}$. The best performance is achieved at $\lambda = 0.01$, where the dense reconstruction guidance enhances backbone representation while maintaining a strong detection signal. This suggests that a moderate weighting allows the two tasks to reinforce each other effectively.
>
> **(3) Large $\lambda$: domination of reconstruction loss.**
>
> When $\lambda$ exceeds 0.05, the training becomes increasingly biased toward the reconstruction objective. Since reconstruction is an easier optimization target, the network tends to overfit to pixel-level restoration at the expense of learning discriminative detection features, ultimately degrading AP. When $\lambda \ge 0.5$, the gradient scale becomes severely imbalanced, making the training process difficult to converge. This effect is more pronounced in our setting because detectors are trained from scratch and are thus more vulnerable to being overwhelmed by dense losses at early stages.
>
> Overall, these observations align with the design goal of our framework: the reconstruction task should complement the detection objective. We have added this analysis to the revised manuscript and appreciate the reviewer for pointing out this important aspect. This suggestion helped us clarify the interaction between the two tasks and improve the completeness of our framework analysis.

---

> ### Author Response · Authors · 2025-11-21
> **Response to Reviewer rmxc (3/3)**
>
> ### **3. Justification for Choosing the Serial $\mathcal{E+D}$ Strategy Over a Parallel Alternative**
>
> As the reviewer noted, Section 3.3 adopts a serial strategy in which enhancement is followed by denoising. The reviewer’s suggestion of a parallel alternative, namely fusing the “enhance-only” result $\mathcal{E(x)}$ with the “denoise-only” result $\mathcal{D(x)}$, is indeed a reasonable design choice. However, we believe that the serial $\mathcal{D(E(x))}$ strategy is better aligned with the characteristics of low-light images and with the goal of constructing an effective fusion target. We summarize our reasoning below.
>
> **(1) Denoising is more stable when illumination is corrected first**
>
> Low-light images exhibit extremely low SNR; object boundaries and textures are often entangled with noise, making it difficult for a denoiser to distinguish true structure from noise when operating directly on the original input $x$. This often leads to over-smoothing of fine structures, or incomplete noise suppression that leaves residual noise in both amplitude and phase.
>
> By first applying the enhancer $\mathcal{E}$, we generate an image $x^{\mathcal{E}}$ where global illumination and contrast are restored. Applying the denoiser to this re-lit image allows it to focus on remaining high-frequency noise rather than struggling to interpret poorly illuminated structures. This produces a more reliable amplitude component for subsequent fusion.
>
> **(2) Serial $\mathcal{D(E(x))}$ aligns naturally with the intended amplitude–phase role separation**
>
> Our Fourier fusion relies on the following division of roles:
>
> - Phase $\mathcal{P(E(x))}$ preserves the structural and semantic information of the enhanced image.
> - Amplitude $\mathcal{A(D(E(x)))}$ maintains the enhanced illumination and style while removing the noise amplified during enhancement.
>
> If the denoiser were applied directly to the raw input to produce $\mathcal{D(x)}$, the amplitude $\mathcal{A(D(x))}$ would still carry the dark, non-uniform illumination style characteristic of low-light images.
>
> Fusing this amplitude with the re-lit structure in $\mathcal{P(E(x))}$ would result in a domain mismatch: the phase reflects enhanced structure, whereas the amplitude remains in the original low-light domain.
>
> In contrast, with the serial strategy both $\mathcal{P}(x^{\mathcal{E}})$ and $\mathcal{A}(x^{\mathcal{E+D}})$ belong to the same re-lit domain, differing only in noise level. This produces a fused target whose structure matches $\mathcal{E(x)}$ while its noise level matches $\mathcal{D(E(x))}$. Section 4.5’s t-SNE analysis confirms this behavior empirically.
>
> **(3) Serial fusion is more stable during training**
>
> The serial design is also advantageous for training stability. In a parallel strategy, the enhancer $\mathcal{E}$ and denoiser $\mathcal{D}$ operate on different input domains (original dark vs. enhanced), which can introduce artifacts during Fourier fusion and may require additional normalization or compensation to match their dynamic ranges and frequency characteristics. In contrast, our $\mathcal{E+D}$ structure removes noise while staying entirely within the same enhanced domain, allowing the simple channel-wise FFT–iFFT procedure to produce stable and natural fused targets without extra adjustments.
>
> In summary, our choice of the serial strategy is based on the observation that (i) denoising is more reliable when applied after illumination correction in low-light conditions, (ii) aligning phase and amplitude within a single re-lit domain naturally supports the intended separation of structural and noise-related information in the fusion process, and (iii) the serial formulation produces more stable target image during training compared to the parallel alternative. We hope this addresses the reviewer’s question regarding the choice of the serial strategy.
>
> We appreciate the reviewer’s insightful suggestions, which led us to quantify the training-time overhead, include a thorough sensitivity analysis of the balancing hyperparameter, and clarify the motivation behind the serial fusion formulation. In the revised version, we will refine the exposition of these design choices to improve clarity and completeness. We thank the reviewer again for the helpful feedback and for recognizing the strengths and practical value of our framework.
>
> **References**
>
> [1] Li, Chongyi, et al. “Embedding Fourier for Ultra-High-Definition Low-Light Image Enhancement.” *The Eleventh International Conference on Learning Representations*. 2023.

---

### Official Review · Reviewer_pa43 · 2025-10-31

**Soundness:** 2
**Presentation:** 3
**Contribution:** 2
**Rating:** 4
**Confidence:** 3

**Summary:**

This paper proposes a self-guided low-light object detection framework to improve detection performance under challenging lighting. The method utilizes image enhancement and denoising, and then the outputs are fused in the Fourier domain. The fusion is used to generate a dense pixel-wise supervision signal encouraging the detector's backbone to learn more robust low-light representations. Experiments are performed on DARK FACE, ExDark, and nuImages datasets, along with ablation and qualitative analyses.

**Strengths:**

1. Fourier-domain fusion: The method’s technical core by combining the amplitude from a denoised image with the phase from an enhanced image using FFT/iFFT is grounded in signal processing principles and attempts to both preserve structure and suppress noise.
2. Extensive evaluation: The experiments cover multiple datasets (DARK FACE, ExDark, nuImages) and detectors. Ablation studies isolate the contributions of each module, and Figure 4 offers qualitative insights.
3. No inference overhead: The method is attractive for applications as it adds no complexity or latency at inference.

**Weaknesses:**

1. While the use of Fourier fusion is motivated by separation of amplitude and phase, the theoretical justifications for this separation, particularly in the context of low-light image statistics and deep feature learning, are not well explained and discussed.
2. The mathematical details for the Fourier fusion are sometimes ambiguous, e.g., the precise computation of bi-level amplitude-phase combination per-channel, whether normalization occurs before/after fusion, whether channel alignment causes artifacts.
3. The impact of severe boundary artifacts, non-Gaussian noise, or extreme illumination imbalance on the auxiliary fused target and final detector features is not explored.

**Questions:**

1. Did you analyze how signal-dependent noise, common in low-light shots, is distributed between the amplitude and phase components?
2. Did you investigate whether the fusion process introduces frequency-domain artifacts, and if so, how were they handled?
3. How does the proposed method perform when significant boundary artifacts are present in the input images?

---

> ### Author Response · Authors · 2025-11-21
> **Response to Reviewer pa43 (1/4)**
>
> We thank Reviewer pa43 for the thoughtful and detailed feedback regarding the theoretical motivation, implementation details, and robustness of our Fourier-domain fusion strategy. The reviewer’s comments helped us identify areas where additional clarification and empirical analysis would substantially strengthen the paper. Below we address each weakness and question in detail, and we will integrate the corresponding clarifications into the revised manuscript.
>
> ### **1. On the Theoretical Justification of Amplitude–Phase Separation in the Fourier Domain**
>
> As the reviewer correctly points out, the motivation for separating amplitude and phase in the Fourier domain to disentangle structural information from noise/brightness factors requires a clearer theoretical grounding. Below, we elaborate on this connection by linking prior findings with our analysis.
>
> Classical vision studies [1] have demonstrated that, in natural images, phase components primarily encode structural and semantic information, whereas amplitude components correspond to texture, contrast, and stylistic properties. Later, Xu et al. [2] revisited this observation in the context of modern deep learning and empirically confirmed: (i) manipulating phase significantly alters object structure and semantics while largely preserving style, and (ii) manipulating amplitude leads to pronounced changes in color, brightness, and style while maintaining object identity. They also showed that phase-only reconstruction restores semantic structure, whereas amplitude-only reconstruction retains style/brightness but becomes nearly unusable for recognition.
>
> More recently, Li et al. [3] extended this analysis to low-light enhancement and reported that brightness and noise predominantly reside in the amplitude, while the structural integrity of objects is relatively well preserved in the phase. Their study also confirmed that low-light–specific degradations (e.g., underexposure, sensor noise) manifest largely as amplitude distortions.
>
> Building upon these three lines of evidence [1–3], our work adopts a new strategy that separates the structurally informative yet noise-amplified phase from the amplitude of the enhanced–denoised image. Concretely, enhancement-only images contain strong structural cues but exhibit substantial noise amplification. Conversely, denoised-after-enhancement images achieve desirable noise levels but inevitably lose boundary fidelity and fine structures. Our approach preserves the phase of the enhanced image while replacing its amplitude with that of the denoised output, thereby compensating for the shortcomings of both.
>
> Figure 2 illustrates this effect: in the red box, the $\mathcal{E+D}$ image shows noticeable structural attenuation, whereas the fused result preserves these structures; in the blue box, the fused image exhibits significantly reduced noise relative to the $\mathcal{E}$-only image. Thus, our Fourier fusion not only aligns with prior analyses but also contributes a new, practical mechanism for integrating enhancement and denoising to benefit downstream low-light detection.
>
> Furthermore, Section 4.5 and Figure 6 analyze the fused target via t-SNE using HOG-based low-level descriptors. As reported, the fused image’s structural/semantic distribution aligns closely with the enhanced-only images (Fig. 6(a)), while its noise-level distribution aligns with enhanced–denoised images (Fig. 6(b)). This provides empirical evidence that our fusion design behaves as intended on real-world low-light datasets.
>
> We will revise the manuscript to incorporate the above theoretical motivations in a more explicit and accessible manner, drawing clearer connections to existing literature on amplitude–phase separation under low-light conditions.

---

> ### Author Response · Authors · 2025-11-21
> **Response to Reviewer pa43 (2/4)**
>
> ### **2. On Fusion Formulation, Normalization, Channel Processing, and Potential Artifacts**
>
> We appreciate the reviewer’s comment that the mathematical specification and implementation details of the Fourier fusion warrant clearer exposition. Below, we provide additional details to ensure reproducibility.
>
> RGB images are first normalized to the [0, 1] range. FFT is then applied independently to each channel. For a single-channel image $x$, its FFT output $X(u,v)$ is decomposed into amplitude and phase as described in Eq. (2) and (3) of the main paper:
>
> $\mathcal{A}(u, v) = \sqrt{R(X(u, v))^2 + I(X(u, v))^2}$
>
> $\mathcal{P}(u, v) = \tan^{-1} \left(\frac{I(X(u, v))}{R(X(u, v))} \right)$
>
> Here, $R(x)$ and $I(x)$ denote the real and imaginary components, and this decomposition is carried out independently for each of the R/G/B channels.
>
> During fusion, we take the enhanced image $x^{\mathcal{E}} = \mathcal{E}(x)$ and its denoised counterpart $x^{\mathcal{E+D}} = \mathcal{D}(x^{\mathcal{E}})$. After FFT decomposition, we combine amplitude from the denoised result with phase from the enhanced result, followed by inverse FFT:
>
> $\hat{x} = iFFT( \mathcal{A}(x^{\mathcal{E+D}}) \cdot e^{j\mathcal{P}(x^{\mathcal{E}})})$
>
> No cross-channel normalization or channel-mixing operations are performed; after iFFT, we then clip each channel to the [0, 1] range before using the result as the decoder’s supervision target.
>
> We also evaluated potential artifacts and channel misalignment. Because we adopt channel-independent FFT, consistent with prior work such as Fourier-DG [2], we do not introduce inter-channel inconsistencies. While Fourier-domain manipulation can sometimes cause ringing or block artifacts, these typically arise when amplitude/phase from unrelated images or spatially mismatched patches are mixed. Our method fuses two images derived from the same enhanced lighting condition input, without patchwise spectral mixing or aggressive spectrum editing. In practice, we did not observe noticeable artifacts. Figure 2 shows that the fused outputs are visually natural and suitable for supervision.
>
> We will include these clarified formulations and implementation details in the revised manuscript to ensure transparency and reproducibility.

---

> ### Author Response · Authors · 2025-11-21
> **Response to Reviewer pa43 (3/4)**
>
> ### **3. On Signal-Dependent Noise, Amplitude–Phase Distribution of Noise, and Noise Modeling**
>
> The reviewer is correct that real low-light noise is signal-dependent, non-Gaussian, and influenced by sensor and ISP characteristics [4]. For this reason, our framework avoids assuming any analytic noise model and instead employs a self-supervised blind-spot denoiser (SDAP) for the denoising module $\mathcal{D}$.
>
> A full theoretical decomposition of signal-dependent noise into amplitude/phase components is beyond the scope of this work. However, existing empirical findings [1–3], together with our t-SNE analysis (Section 4.5), consistently show that noise component concentrates mainly in the amplitude, while phase remains more stable and structure-preserving.
>
> Additionally, our denoiser ablation (Table D–I) further supports this design choice:
>
> | Method | Enhancer | Denoiser | AP |
> | --- | --- | --- | --- |
> | D | SCI | Median blur | 73.6 |
> | E | SCI | Restormer (σ=15) | 75.0 |
> | F | SCI | Restormer (σ=25) | 73.0 |
> | G | SCI | Restormer (σ=50) | 73.0 |
> | H | SCI | Restormer (real-noise) | 76.0 |
> | I | SCI | SDAP | 76.6 |
>
> Gaussian-based denoisers (E–G) depend heavily on the assumed noise level σ; mismatches cause performance degradation even below simple median filtering (D). This confirms that Gaussian noise assumptions poorly match real low-light noise. When we train the same Restormer architecture on a real-noise dataset [6] (H), its performance improves over the Gaussian-trained variants, indicating that using more realistic noise statistics leads to more effective suppression of real-world noise. SDAP (I), while also targeting real-world noise, is self-supervised and can be trained directly on the target low-light dataset, which allows it to better adapt to the actual signal-dependent noise characteristics in our setting. Its superior AP (76.6) suggests that adopting a self-supervised denoiser that learns from target-domain noise, rather than relying on a fixed parametric noise model, is an effective choice within our framework.
>
> In summary, our hypothesis, which supported by [1–3] and validated through t-SNE and ablations, is that noise predominantly affects amplitude, whereas structure is captured in phase. We will clarify this analysis and expand discussion of signal-dependent noise in the revision.

---

> ### Author Response · Authors · 2025-11-21
> **Response to Reviewer pa43 (4/4)**
>
> ### **4. On Boundary Artifacts and Performance Under Extreme Illumination Imbalance**
>
> Finally, we address the reviewer’s question regarding the influence of boundary artifacts, non-Gaussian noise, and extreme illumination imbalance on the fused target and the detector’s features.
>
> In this work, we evaluate these issues using the nuImages nighttime driving benchmark, which naturally contains such challenging phenomena. The nighttime scenes in nuImages frequently exhibit intense headlight glare, high-frequency noise along road boundaries, and strong boundary discontinuities and illumination imbalance between objects and background. As shown in Section 4.4 and Figure 5, our framework demonstrates robustness in these conditions, achieving +2.9 mAP when trained from scratch and +2.6 mAP when finetuned after daytime pretraining (YOLOv8-m).
>
> Importantly, enhancement-based methods and domain adaptation approaches (e.g., MAET, DAI-Net) either degrade or provide only limited improvements under this setting, where boundary glare and huge domain gaps coexist. In contrast, our method continues to provide consistent gains even in real nighttime driving scenarios with severe boundary artifacts.
>
> Thus, the nuImages experiment provides indirect yet practical empirical evidence, using real-world nighttime driving data, that our framework remains robust to boundary artifacts and extreme illumination imbalance.
>
> We appreciate the reviewer’s careful assessment and constructive questions, which guided us to deepen the theoretical explanation of amplitude–phase separation, clarify the fusion formulation, and better articulate the robustness of our approach under real-world low-light degradations. In the revision, we will refine the presentation of Fourier fusion details, and expand the discussion to ensure the method’s motivation and behavior are accessible and reproducible. We thank the reviewer again for the valuable insights, which have helped us significantly strengthen the clarity and completeness of the paper.
>
> **References**
>
> [1] Piotrowski, Leon N., and Fergus W. Campbell. “A demonstration of the visual importance and flexibility of spatial-frequency amplitude and phase.” *Perception* 11.3 (1982): 337–346.
>
> [2] Xu, Qinwei, et al. “A fourier-based framework for domain generalization.” *Proceedings of the IEEE/CVF Conference on Computer Vision and Pattern Recognition*. 2021.
>
> [3] Li, Chongyi, et al. “Embedding Fourier for Ultra-High-Definition Low-Light Image Enhancement.” *The Eleventh International Conference on Learning Representations*. 2023.
>
> [4] Lee, Wooseok, Sanghyun Son, and Kyoung Mu Lee. “AP-BSN: Self-supervised denoising for real-world images via asymmetric PD and blind-spot network.” *Proceedings of the IEEE/CVF Conference on Computer Vision and Pattern Recognition*. 2022.
>
> [5] Zamir, Syed Waqas, et al. “Restormer: Efficient transformer for high-resolution image restoration.” *Proceedings of the IEEE/CVF Conference on Computer Vision and Pattern Recognition*. 2022.
>
> [6] Abdelhamed, Abdelrahman, Stephen Lin, and Michael S. Brown. "A high-quality denoising dataset for smartphone cameras." *Proceedings of the IEEE conference on computer vision and pattern recognition*. 2018.

---

> > ### Comment · Reviewer_pa43 · 2025-11-25
> >
> > Thanks for your efforts to address my concerns. I'm pleased to raise my score.

---

### Official Review · Reviewer_Y1L8 · 2025-10-31

**Soundness:** 3
**Presentation:** 3
**Contribution:** 3
**Rating:** 6
**Confidence:** 4

**Summary:**

The paper introduces a novel framework to improve object detection in low-light conditions. The core idea is to use a detachable auxiliary pipeline during the training phase to provide a self-guided supervisory signal to the main detector's backbone. This pipeline, which is removed at inference time, consists of self-supervised image enhancement and denoising modules whose outputs are combined in the Fourier domain to create a high-quality target image. The backbone is then trained on a multi-task loss, combining the detection loss with a reconstruction loss based on this generated target. The authors claim this approach improves feature representation for low-light scenes, achieving state-of-the-art results on several benchmarks (DARK FACE, ExDark, nuImages) without adding any computational cost during inference.

**Strengths:**

+ The manuscript is well-represented and easy-to-follow.

+ The auxiliary pipeline (enhancer, denoiser, fusion) is completely detached after training. This means the detector is identical in architecture, parameters, and speed to the baseline model, yet it performs significantly better.

+ The effectiveness of the framework is validated across three different datasets and with multiple detector architectures.

**Weaknesses:**

+ The paper claims that performance stems from the framework design itself. However, in Table 5, using simple modules (Gamma Correction, Gaussian Blur) yields a 70.9 mAP, while advanced modules (SCI, SDAP) are required to reach the top performance of 76.6 mAP. This indicates the framework's effectiveness is tied to the quality of the chosen enhancer and denoiser. It would be better to see more variant of enhancer/denoiser pairs and more discussion for this.

+ The model is trained with a multi-task loss combining sparse detection (L_det) and dense pixel-reconstruction (L_self). How is the sensitivity to the weighting hyperparameter $\lambda$?

+ Figure 4, Retienxformer -> Retinexformer. The authors should carefully check the manuscript to eliminate typos and grammarical errors.

**Questions:**

Please refer to weakness part.

---

> ### Author Response · Authors · 2025-11-21
> **Response to Reviewer Y1L8 (1/2)**
>
> We thank Reviewer Y1L8 for the insightful feedback, as well as for the positive evaluation of our work’s. The comments, particularly regarding the module dependency, the sensitivity of the weighting hyperparameter, and the manuscript clarity, have been highly valuable in helping us strengthen the paper. Below, we address each point in detail with additional experiments and analysis.
>
> ### **1. Comment on Module-Dependency and Framework Generality**
>
> We thank the reviewer for the positive assessment of our contributions and for raising this insightful question regarding the dependency of our framework on the choice of enhancer–denoiser modules. To address this point more thoroughly, we conducted additional experiments using various combinations of enhancers and denoisers, and we provide the results and analysis below.
>
> **(1) Effect of Enhancer Choice (Methods A–C)**
>
> With the denoiser fixed as SDAP, we compared three enhancers:
>
> | Method | Enhancer | Denoiser | AP |
> | --- | --- | --- | --- |
> | A | Retinexformer | SDAP | 72.5 |
> | B | Zero-DCE | SDAP | 73.8 |
> | C | SCI | SDAP | 76.6 |
>
> Retinexformer is a supervised enhancer requiring ground-truth images, which makes it difficult to adapt to target datasets such as DARK FACE that lack paired illumination labels. Zero-DCE and SCI, in contrast, are self-supervised methods and can be trained directly on the target dataset. Both outperform the supervised enhancer (A), supporting our core claim that the proposed self-guided framework is naturally aligned with enhancement modules that do not rely on ground truth. Additionally, SCI (C), the more advanced of the two self-supervised methods, achieves the highest AP, indicating that better enhancement quality leads to better supervisory signals during training.
>
> Importantly, these results do not contradict our framework’s contribution; rather, they highlight that the auxiliary branch functions as a flexible component whose benefits scale with the strength of the chosen enhancer.
>
> **(2) Effect of Denoiser Choice (Methods D–I)**
>
> Next, with the enhancer fixed as SCI, we varied the denoiser:
>
> | Method | Enhancer | Denoiser | AP |
> | --- | --- | --- | --- |
> | D | SCI | Median blur | 73.6 |
> | E | SCI | Restormer (σ=15) | 75.0 |
> | F | SCI | Restormer (σ=25) | 73.0 |
> | G | SCI | Restormer (σ=50) | 73.0 |
> | H | SCI | Restormer (real-noise) | 76.0 |
> | I | SCI | SDAP | 76.6 |
>
> Median blur (D) represents a naive rule-based denoiser, and as expected, it offers limited denoising capability. Restormer[1]-based models (E–H), which share the same architecture but differ only in the noise assumptions used during training, allow us to examine how denoising performance varies purely as a function of the underlying noise model. The Gaussian-trained variants (E–G) show large performance differences depending on the assumed σ. When σ is mismatched with real-world noise conditions—as in (F) and (G)—performance can drop below that of the simple median blur. The Restormer trained on real-world noise data[2] (H) performs more robustly than its Gaussian counterparts, indicating that more realistic noise assumptions indeed help stabilize the amplitude component used in fusion. SDAP (I), while also targeting real-world noise, is self-supervised and can be trained directly on the target dataset, which appears to offer additional adaptability to the noise characteristics of the specific low-light domain. As a result, SDAP achieves the highest performance in this ablation.
>
> Overall, these extended results further support our main claim. While it is natural that more advanced enhancement or denoising modules lead to higher detection performance, the improvements relative to the baseline detector remain consistently substantial across all module choices (including simple configurations such as gamma correction or median blur presented in Table 5). This consistent improvement across diverse setups suggests that the key contribution of our work lies in the framework design itself: the detachable auxiliary pathway provides an effective form of self-guided, dense supervision that enhances low-light feature learning regardless of the specific choice of modules. We appreciate the reviewer’s suggestion and will clarify this point and integrate the additional results and analysis in the revised manuscript.

---

> ### Author Response · Authors · 2025-11-21
> **Response to Reviewer Y1L8 (2/2)**
>
> ### **2. Sensitivity analysis of hyperparameter** $\lambda$
>
> We thank both Reviewer Y1L8 and Reviewer rmxc for raising the importance of analyzing the weighting factor between the detection loss $L_{\text{det}}$ and the self-guided reconstruction loss $L_{\text{self}}$. Since our framework jointly optimizes sparse detection supervision and dense pixel-level guidance, understanding the sensitivity to the hyperparameter $\lambda$ is indeed crucial. In response, we conducted an extensive study across a wide range of $\lambda$ values, and the results are summarized in the table below.
>
> | Hyperparameter $\lambda$ | AP |
> | --- | --- |
> | 0.0001 | 72.5 |
> | 0.0010 | 74.0 |
> | 0.0100 | **76.6** |
> | 0.0500 | 67.3 |
> | 0.1000 | 61.6 |
> | 0.5000 | Did not converge |
> | 1.0000 | Did not converge |
>
> Several trends can be observed from the results:
>
> **(1) Small $\lambda$: insufficient dense supervision.**
>
> When $\lambda$ is too small, the influence of $L_{\text{self}}$ becomes marginal. Although the backbone still benefits from a small amount of dense guidance, the improvement remains limited, as observed at $\lambda \le 10^{-3}$.
>
> **(2) Moderate $\lambda$: optimal balance.**
>
> We observe performance improvements as $\lambda$ increases from $10^{-4}$ to $10^{-2}$. The best performance is achieved at $\lambda = 0.01$, where the dense reconstruction guidance enhances backbone representation while maintaining a strong detection signal. This suggests that a moderate weighting allows the two tasks to reinforce each other effectively.
>
> **(3) Large $\lambda$: domination of reconstruction loss.**
>
> When $\lambda$ exceeds 0.05, the training becomes increasingly biased toward the reconstruction objective. Since reconstruction is an easier optimization target, the network tends to overfit to pixel-level restoration at the expense of learning discriminative detection features, ultimately degrading AP. When $\lambda \ge 0.5$, the gradient scale becomes severely imbalanced, making the training process difficult to converge. This effect is more pronounced in our setting because detectors are trained from scratch and are thus more vulnerable to being overwhelmed by dense losses at early stages.
>
> Overall, these observations align with the design goal of our framework: the reconstruction task should complement the detection objective. We have added this analysis to the revised manuscript and appreciate the reviewer for pointing out this important aspect. This suggestion helped us clarify the interaction between the two tasks and improve the completeness of our framework analysis.
>
> We appreciate the reviewer’s thoughtful suggestions, which prompted us to perform additional analyses and improve the clarity of our contributions. In the revised manuscript, we will fix the noted issues (e.g., “Retienxformer → Retinexformer” in Figure 4). We hope that the added experiments and clarifications directly address the reviewer’s concerns and further substantiate the robustness and generality of our framework. Thank you again for the valuable feedback.
>
> **References**
>
> [1] Zamir, Syed Waqas, et al. "Restormer: Efficient transformer for high-resolution image restoration." *Proceedings of the IEEE/CVF conference on computer vision and pattern recognition*. 2022.
>
> [2] Abdelhamed, Abdelrahman, Stephen Lin, and Michael S. Brown. "A high-quality denoising dataset for smartphone cameras." *Proceedings of the IEEE conference on computer vision and pattern recognition*. 2018.

---

> > ### Comment · Reviewer_Y1L8 · 2025-11-25
> >
> > Thanks for your comprehensive response. It alleivates my concern. I'd like to keep my recommendation for accepting this paper.

---

### Comment · Area_Chair_yfJm · 2025-11-22

Dear Reviewers,

How do you think of the rebuttal?

---

### Author Response · Authors · 2025-12-04
**Author Final Remarks**

We sincerely thank the AC and all reviewers for their careful evaluation, constructive feedback, and thoughtful discussions. We appreciate the recognition of our work’s technical novelty, practical value, and consistent performance gains across multiple low-light detection benchmarks.

---

### **Key Contributions**

Our work introduces a **Self-Guided Low-Light Object Detection Framework** that enhances feature learning for low-light scenarios without modifying the detector at inference time. The key contributions are:

- **Detachable auxiliary pipeline:**

    We design an enhancement–denoising–fusion pipeline that provides a dense supervisory target *only during training*. The detector architecture, parameters, and runtime remain exactly identical to the baseline model at inference.

- **Fourier-domain amplitude–phase fusion:**

    We leverage amplitude–phase separation to combine the structure-preserving phase of the enhanced image with the noise-suppressed amplitude of the denoised image, yielding a high-quality self-guiding signal grounded in well-established signal-processing principles.

- **Self-supervised modules aligned with low-light statistics:**

    Both enhancer and denoiser operate without ground-truth well-lit or clean labels. This design naturally adapts to real low-light noise/illumination characteristics and generalizes across DARK FACE, ExDark, and nuImages.

- **Strong, consistent empirical improvements:**

    Our method achieves significant gains across three benchmarks and multiple detector families while adding *zero inference overhead*, demonstrating both practical utility and reproducibility.


---

### **Concerns Addressed**

In the discussion phase, we conducted additional experiments, added analyses, clarified technical details, and corrected manuscript issues. Key responses include:

**(1) Module dependency & generality**

We performed extensive ablations across multiple modules. The auxiliary branch yields substantial gains over the baseline even with simple modules, while more advanced self-supervised enhancers and realistic denoisers further boost AP, highlighting that the framework is flexible and that its benefits scale with module quality.

**(2) Sensitivity of $\lambda$ (loss weighting)**

We added a $\lambda$-sweep. Moderate $\lambda$ (=0.01) achieves best performance. Too-small $\lambda$ gives weak guidance, too-large $\lambda$ overwhelms detection gradients. This clarifies the complementary nature of sparse/dense supervision.

**(3) Theoretical and implementation clarity of Fourier fusion**

We expanded justification for amplitude–phase separation, detailed the per-channel FFT/iFFT formulation, normalization, and artifact considerations, and connected to findings from prior signal-processing and low-light studies.

**(4) Noise characteristics and boundary artifacts**

We analyzed noise assumptions through denoiser ablations, t-SNE distributions, and empirically validated robustness through nuImages experiments, where real nighttime scenes exhibit strong glare, boundary artifacts, and illumination imbalance.

**(5) Serial vs. parallel pipeline design**

We provided rationale showing that serial pipeline avoids domain mismatch, yields more stable amplitude–phase alignment, and produces more reliable fused targets.

**(6) Training-time cost**

We measured and reported the training overhead (~1.8x), clarified that the cost is one-time, and emphasized that inference remains unchanged.

---

### **Discussion Summary**

We appreciate the reviewers engagement during the discussion phase. Reflecting the newly added analyses and clarifications:

- **Reviewer Y1L8** confirmed that concerns were resolved and maintained a positive recommendation.
- **Reviewer pa43** updated the evaluation, raising the Soundness from **2 to 3**, the Contribution from **2 to 3**, and the overall rating from **4 to 6**, noting that our responses satisfactorily addressed the concerns.

The revision incorporates the additional analyses, clarified explanations, and manuscript refinements suggested during the discussion.

---

We believe our framework offers a practical solution to low-light object detection by leveraging frequency-domain guidance and self-supervised modules without any inference-time cost. We hope this work contributes meaningfully to robust perception under challenging illumination.

Thank you again for your time, careful assessment, and constructive feedback throughout the review process.

---

### Meta-Review · Area_Chair_Q6u4 · 2026-01-07

**Summary:**

This paper introduces a training-only auxiliary enhancement, denoising and Fourier fusion pipeline that improves low-light object detection without changing inference-time architecture or cost. The rebuttal provides additional ablations, hyperparameter sensitivity analysis, training cost measurements and clarified explanations, which address the main concerns. Experiments show consistent gains across benchmarks, and reviewer opinions converged more positively after discussion. While some limitations remain, including reliance on auxiliary modules and increased training time, the work is technically sound and practically useful, and I consider it suitable for acceptance.

**Reviewer Concerns:**

Reviewer Y1L8 raised concerns about whether the framework’s effectiveness depended too strongly on the choice of enhancement and denoising modules, as well as the lack of analysis on the loss weighting hyperparameter. The rebuttal provided extensive ablations and a full λ sweep that directly addressed both points, and Y1L8 acknowledged that these concerns were resolved.

Reviewer pa43 questioned the theoretical motivation behind the Fourier-domain fusion strategy, the precise mathematical formulation, and whether noise and boundary artifacts would lead to failure cases. The authors responded with detailed justification supported by literature, full implementation clarifications, and empirical validation on nuImages, and pa43 stated satisfaction with the revisions.

Reviewer rmxc pointed out two missing elements: explicit reporting of the training-time cost and justification for choosing a serial fusion pipeline instead of a parallel one. The authors quantified the training computation overhead and presented reasoning grounded in illumination statistics and empirical behavior showing better stability with the serial design; rmxc indicated that these responses sufficiently addressed their questions.

The remaining points that could be considered partially open, though not fatal, are that performance still scales with auxiliary module strength (noted by Y1L8) and that fusion design remains primarily grounded in empirical observations rather than formal theory (originally highlighted by pa43). These aspects do not contradict the paper’s claims but suggest future explorations.

**Reviewer Scores:**

Reviewer Y1L8 initially gave a positive assessment of score 6, and explicitly stated after rebuttal that the concerns were alleviated.

Reviewer pa43 initially gave a 4 due to missing theoretical depth and implementation clarity; after the rebuttal, the reviewer raised the score.

Reviewer rmxc initially gave score 6 while requesting clarification on training cost and loss weighting; following the detailed response, rmxc confirmed that their concerns were addressed.

The fourth assigned reviewer withdrew early and did not contribute, so no score adjustment applies.

---

### Decision · Program_Chairs · 2026-01-26

Accept (Poster)